# Case Reports of Pregnancy-Related Cerebral Venous Thrombosis in the Neurology Department of the Emergency Clinical Hospital in Constanta

**DOI:** 10.3390/life12010090

**Published:** 2022-01-09

**Authors:** Any Docu Axelerad, Lavinia Alexandra Zlotea, Carmen Adella Sirbu, Alina Zorina Stroe, Silviu Docu Axelerad, Simona Claudia Cambrea, Lavinia Florenta Muja

**Affiliations:** 1Department of Neurology, General Medicine Faculty, Ovidius University, 900470 Constanta, Romania; axelerad.docu@365.univ-ovidius.ro (A.D.A.); laviniamuja@gmail.com (L.F.M.); 2Radiology-Medical Imaging, Sf Apostol Andrei Emergency County Clinical Hospital, 900591 Constanta, Romania; laviniazlotea23@gmail.com; 3Department of Neurology, Titu Maiorescu University, 040441 Bucharest, Romania; carmen.sirbu@prof.utm.ro; 4Faculty of General Medicine, ‘Vasile Goldis’ University, 317046 Arad, Romania; docu.silviu@yahoo.com; 5Infectious Diseases Clinic, Clinical Infectious Diseases Hospital, 900708 Constanta, Romania; claudia.cambrea@univ-ovidius.ro; 6Department of Infectious Diseases, General Medicine Faculty, Ovidius University, 900470 Constanta, Romania

**Keywords:** cerebral venous thrombosis, pregnancy, puerperium, voluntary abortion

## Abstract

Cerebral venous thrombosis accounts for 0.5–1% of all cerebrovascular events and is one type of stroke that affects the veins and cerebral sinuses. Females are more affected than males, as they may have risk factors, such as pregnancy, first period after pregnancy, treatment with oral contraceptives treatment with hormonal replacement, or hereditary thrombophilia. This neurological pathology may endanger a patient’s life. However, it must be suspected in its acute phase, when it presents with variable clinical characteristics, so that special treatment can be initiated to achieve a favorable outcome with partial or complete functional recovery. The case study describes the data and the treatment of two patients with confirmed cerebral venous thrombosis with various localizations and associated risk factors, who were admitted to the neurology department of the Sf. Apostol Andrei Emergency Hospital in Constanta. The first patient was 40 years old and affected by sigmoid sinus and right lateral sinus thrombosis, inferior sagittal sinus, and right sinus thrombosis, associated with right temporal subacute cortical and subcortical hemorrhage, which appeared following a voluntary abortion. The second case was a patient aged 25 who was affected by left parietal cortical vein thrombosis, associated with ipsilateral superior parietal subcortical venous infarction, which appeared following labor. The data are strictly observational and offer a perspective on clinical manifestations and clinical and paraclinical investigations, including the treatment of young patients who had been diagnosed with cerebral venous thrombosis and admitted to the neurology department.

## 1. Introduction

Cerebral venous thrombosis (CVT) is an uncommon type of neurological pathology. It is an embedded cerebrovascular disease, may appear regardless of age, and corresponds to 0.5–1% of all cerebral and vascular strokes [1]. It presents a remarkable range of symptoms and signs, the most frequent being cephalalgia, convulsions, focal neurological deficits, papillary edema, and alternating general health status and consciousness [2].

Around 20% of CVST-related strokes have been documented in young Asian females [3]. Bano et al. found that, in 68.8% of patients, the gender-specific pathogenesis of CVST was related to pregnancy and the postpartum [4]. CVT occurs in 2–57% of pregnancy-related strokes, with the majority of instances occurring during the postpartum period [5]. In Bajko et al.’s [6] research, females comprised around 60% of the study group, and a significant proportion of the female patient population was in the puerperium (17.2%).

Modifications of the cerebral parenchyma appear in cerebral venous thrombosis due to an obstruction in the venous system and intracranial sinuses. The primary locations of flow blockages appear to be at the intersections of cerebral veins and major cerebral sinuses, with a preference for the superior sagittal and transverse sinuses [7,8].

In patients with this neurological pathology, extremely diverse symptomatology may be encountered, both at the beginning and during its evolution, and younger patients can suffer from it. Female patients are particularly affected when they have risk factors such as pregnancy, puerperium, use of oral contraceptives or hormonal replacement therapy; patients with hereditary thrombophilia may also be affected [9].

Pregnancy-related stroke develops in 30.0 out of every 100,000 pregnancies; hemorrhagic and ischemic strokes happen in two thirds of cases, and CVST occurs in one third [10]. The first postpartum month, the perinatal period and the third trimester of pregnancy are the most strongly associated with the etiology of CVST, although just a few instances of CVST have been documented in early pregnancy. Our patients had pregnancy-related situations: the first patient had an abortion via curettage and the second patient was in the sixth day of puerperium after a caesarian delivery.

As a rare high-risk disease, cerebral venous thrombosis must be suspected in cases accompanied by cephalalgia and other focal neurological deficits, altered consciousness, and convulsions, even in the acute phase when the disease has variable symptoms [11].

The diagnosis of CVST should be confirmed by clinical and paraclinical manifestations, the associated risk factors, and a neurological examination. The imaging investigations are extremely important for confirming the diagnosis and the most reliable imaging type is nuclear magnetic resonance (NMR).

The most useful methods for establishing a diagnosis of CVT are imaging investigations, and the most precise method involves a combination of magnetic resonance imaging and angiography. All patients suspected of having CVT should undergo screening for prothrombotic states, along with an evaluation of the levels of antithrombin III, levels of S and C proteins, homocysteine levels, mutations of prothrombin levels, and levels of antiphospholipid antibodies [12].

The thrombophilia test is useful for people with CVST-associated risk factors who have a history of thromboembolism. In particular, it is beneficial and essential for pregnant women with another associated risk factor.

Treatment aims to halt the progression of a thrombus through the elimination of obstacles in the sinus lumen and the affected veins, thereby improving the prothrombotic status with the purpose of preventing venous thrombotic events, and at the same time, preventing the occurrence of relapses. Moreover, treatment also aims to optimize functional recovery. Drug treatments may consist of blood thinners or thrombolytic medications; the latter are suggested for destroying the blood clots that have formed if the thrombosis is serious.

In this article, we present the cases of two patients with neurological pathology who were admitted to the neurology department of Sf. Apostol Andrei Emergency Hospital of Constanta and who progressed favorably under special treatment. The Ethics Committee for Clinical Studies at the Constanta County Emergency Clinical Hospital (registration number 30/1.11.2021) approved the study, which was conducted in accordance with the Declaration of Helsinki. All subjects gave written informed permission before of the enrolment.

The first patient presented with sigmoid sinus and right lateral sinus thrombosis and inferior sagittal sinus, and right sinus thrombosis, associated with right temporal subacute cortical and subcortical hemorrhage, a pathology that appeared following a voluntary abortion. The second patient was affected by left parietal cortical vein thrombosis associated with an ipsilateral superior parietal subcortical venous infarction that appeared following labor and delivery by a caesarian operation.

## 2. Results

### 2.1. Case 1

The first case was a female patient aged 40, known to be affected by laparoscopic cholecystectomy and abortion by curettage in ambulatory surgery. The patient was ad-mitted to the neurology department for a period of 11 days due to a decrease in muscular strength in the right hemibody and a speech disorder that onset during the same day.

At the emergency ward, the patient underwent a native and contrast-enhanced CT scan, which highlighted the following: the right temporal surface was most probably compatible, in terms of imaging, with encephalitis; a cranial cerebral MRI examination and a pulmonary radiography (X-ray), which did not reveal pleuro-pulmonary evolutive lesions, were recommended.

When admitted to the neurology department, during the neurological objective examination, the patient was conscious, barely cooperative, had expressive aphasia, had no stiffness in the back of the head, had normal ocular motricity, and had no nystagmus; further, the patient presented with right hemiparesis 3/5 equally distributed, and a cutaneous plantar reflex with bilateral flexion.

On the first day of hospitalization, the patient underwent a cerebral native and contrast-enhanced MRI and angiography, and a CT venography, which highlighted sigmoid sinus and right lateral sinus thrombosis, and inferior sagittal sinus and right sinus thrombosis, associated with right temporal cortical and subcortical subacute hemorrhage, supratentorial recent subacute synchronous lacunar infarct, (cytotoxic and vasogenic) thalamic–lenticular–caudal oedema, and supratentorial non-specific demyelinating lesions.

On the first day of hospitalization, biological material was collected in order to create the hereditary thrombosis profile, which was transmitted to the Clinical Service of Patho-logical Anatomy and tested to identify the mutations associated with cardiovascular disease and thrombophilia.

The following genotypes were identified: heterozygous for V factor mutation *H1299R* (R2) and heterozygous for mutation *4G* of *PAI-1*. Moreover, the *A1/A1* haplotype of *EPCR* was identified.

On the second day of admission, the patient underwent native thoracic CT and angiography for pulmonary arteries, without any images suggesting pulmonary thromboembolism.

On the same day (the second day of admission), the patient underwent a gynecological exam. The diagnosis of incomplete abortion was confirmed, and the patient was recommended to perform beta-human chorionic gonadotropin (hCG) after 48 h. The patient was recommended to take antibiotics (2 g intravenous ampicillin every 12 h and 80 g intravenous gentamicin every 12 h), and a nonsteroidal anti-inflammatory drug (75 mg diclofenac, one 0.2 mg tablet every 12 h) and to undergo reevaluation according to the results of the tests. Uterine curettage was recommended to be performed after neurological rebalancing of the patient.

During admission, on the fifth day of hospitalization, the patient underwent a cardiac assessment. She also had a blood pressure of 105/60 mmHg, a heart rate of 64 beats per minute, a normal electrocardiogram (ECG) with sinus rhythm, and angiography of the pulmonary arteries that infirmed pulmonary thromboembolism.

On the seventh day of admission, a gynecological reassessment was requested for the result of beta-hCG testing, and the investigation indicated a decrease in its value. Following a consultation, the patient was advised to continue therapy with antibiotics and to repeat the beta-hCG test after 7 days for a reassessment of the results.

On the ninth day, the gynecological reassessment indicated a paraclinical decrease in beta-hCG and the patient was recommended to repeat this test on a weekly basis until negative results were obtained and to have a gynecological assessment in ambulatory after 1 week. A hematological assessment was also requested on the ninth day of admission, which confirmed that the mutations detected in the hereditary thrombophilia profile fell into the low-risk class. It was recommended that the screening test for thrombophilia be intensified for C and S proteins, AT III, the dosage of serum homocysteine, and lupus anticoagulant in the ambulatory. The details of treating the patient and the findings can be seen in Figure 1.

During her admission, the patient received brain depletive medication (20% mannitol, 250 mL every 12 h, for 5 days), hydro-electrolytic rebalancing (500 mL of normal saline solution every 12 h), heparin with a small molecular weight (0.4 mL enoxaparin sodium, one ampoule per day during the entire period of admission), cortico-steroid anti-inflammatory drugs (one ampoule of intravenous dexamethasone every 12 h for the first 2 days of admission), benzodiazepine (one ampoule of diazepam in 10 mL of a normal intravenous saline solution, given slowly in case of convulsive spasm), antibiotics (2 g intravenous ampicillin every 12 h, 80 mg intravenous gentamicin every 12 h), a non-steroidal anti-inflammatory (one ampoule of ketoprofen per day), a gastric protector (one 40 mg pantoprazole tablet per day), a probiotic enhancer (one 250 mg enterolum tablet twice a day, 2 h after the antibiotic), and an oral anticoagulant (4 mg acenocoumarol) as follows: 1/4 tablet on the seventh day of admission; 1/2 tablet per day on days 8 and 9; 3/4 tablet per day on days 10 and 11, with international normalized ratio (INR) control performed on a daily basis, in order to adjust the dose according to the INR. The patient recovered well.

At discharge, the patient continued the treatment with the oral anticoagulant (4 mg acenocoumarol as follows: 3/4 tablet alternating with 1 tablet per day, according to the INR dosage every 7, 14, and 21 days, then, on a monthly basis), to which a pain-killer was added (one tablet metamizole, if necessary), along with a brain trophic (one 400 mg Actovegin tablet, three times per day) and a dietary supplement with alpha-lipoic acid and a complex of vitamins.

Upon discharge, the neurological examination indicated that the patient was conscious, cooperative, oriented in time and space, without stiffness in the back of the head, no nystagmus, with normal ocular motricity, denial of diplopia, possible deglutition for liquid and solid food, normal speech, no movement deficits, no sensitivity or coordination disorders, and a cutaneous plantar reflex with bilateral flexion. The images of the MRI can be seen below in Figure 2.

### 2.2. Case 2

The second case we present was a 25-year-old female patient, with no pathological personal history, in the sixth day of puerperium after caesarian delivery, who came to the Emergency Unit of the Sf. Apostol Andrei Clinical Hospital in Constanta. The patient presented with cephalalgia, paresthesia of the right hemibody, a decrease in muscular strength in the right upper limb, and tonic–clonic convulsive spasms.

Following medical history, clinical examination and clinical and paraclinical investigations, it was decided that the patient should be admitted to the obstetrics and gynecology department, with the following diagnostics: 6-day puerperium after caesarian delivery, a mild form of secondary anemia and convulsion spasms to be investigated. She was oriented in time and space, was afebrile, had normally colored skin, had post-caesarian operation soft plaque in the process of healing (sutures removed), had a weak muscular system, was hypokinetic in the right upper and lower limbs, had a soft abdomen, and had a contracted uterus, had serosanguinous leaking, had supple breasts, was lactating, had a movement deficit in the right upper limb, with a lingual trauma mark.

The native cerebral CT scan highlighted a normality, with no detectable cranio-cerebral pathological modifications. The patient also underwent a native thorax CT and angiography for the pulmonary arteries, which did not indicate images suggestive of pulmonary thromboembolism.

A neurological consultation was requested when she was admitted to the gynecology department. When consulted, the patient was conscious, cooperative, oriented in time and space, had symmetrical pupils, was reactive on the median line, with a bitten tongue, showed right hemiparesis (upper limb 3/5 and lower limb 4/5) and had a bilaterally traced plantar cutaneous reflex. The diagnosis of tonic–clonic generalized convulsive spasm with tongue biting, and a post-critical status was given.

A cerebral and angiography MRI with venous time and reevaluation of the result was recommended, along with treatment with antiepileptic drugs (200 mg carbamazepine, 1/2 tablet per day; one intravenous ampoule of phenytoin 250 mg/5 mL, diluted in case of convulsive spasm), and a platelet antiaggregatory agent (one 75 mg aspenter tablet per day). Another neurological examination was requested on the same day, as the patient presented involuntary movements of the right lower limb. She received treatment with antiepileptic drugs (the dose was increased to one 200 mg carbamazepine tablet three times per day, plus one ampoule of intravenous phenytoin 250 mg/5 mL in 20 mL of normal saline solution), a cerebral depletive (20% mannitol, one 250 mL ampoule twice a day), a corticosteroid anti-inflammatory (one dexamethasone ampoule twice a day), a gastric protector (one 40 mg pantoprazole tablet per day), and heparin with a small molecular weight (one 0.6 mL enoxaparin sodium ampoule per day). The use of the steroid anti-inflammatory drugs in cerebral venous thrombosis was the choice of the treating physician.

On the second day of hospitalization, the patient underwent a native and contrast-enhanced cerebral MRI and angiography with arterial and venous sequence. The examination was suggestive of the left parietal cortical vein associated with ipsilateral superior parietal subcortical venous infarction and demyelinating lesions organized in the crown radiated and parietal on the right side, most probably with an ischemic vascular sublayer. This was the reason why the patient was transferred to the neurology department on the same day.

When admitted to the neurology department, on the second day of hospitalization, the patient was objectively examined and presented a good general health: a Glasgow Coma Scale (GCS) of 15 points, afebrile, conscious, cooperative, oriented in time and space, no deficit of the cranial nerves, a tongue with a traumatic trace, right hemiparesis 4/5 (predominantly brachial), right tactile hypesthesia; she denied any lost pregnancies, thrombophilia, or eclampsia.

On the third day of admission, biological material was drawn for the purpose of examining the hereditary thromboliphilia profile and was sent to the Clinical Service of Pathological Anatomy to be tested to identify mutations associated with cardiovascular disease and thrombophilia.

The test indicated the following genetic variants: heterozygous double genotype for the *C677T* and *A1298C* alleles in the *MTHFR* gene, mutant homozygous for the *V34L* allele in the *factor XIII* gene, wild type homozygous genotype *5G/5G* of the *PAI-1* gene and haplotype *A2/A2 (H2/H2)* of *EPCR.* The findings about and treatment of the patient can be seen in Figure 3.

During admission to the neurology department, the patient was treated with a cerebral depletive (20% mannitol, 250 mL every 12 h for 5 days), hydro-electrolytic rebalancing (500 mL normal saline solution two times a day), heparin with low molecular weight (one 0.6 mL enoxaparin sodium ampoule every 12 h), painkillers (one metamizole ampoule every 12 h), antiepileptic medicines (one 200 mg carbamazepine tablet three times a day), a corticosteroid anti-inflammatory (dexamethasone, 1/2 ampoule every 12 h), benzodiazepine (one ampoule of intravenous diazepam slowly in 10 mL of normal saline solution, in case of convulsive spasm), a gastric protector (one 40 mg pantoprazole tablet per day), an oral anticoagulant (4 mg acenocoumarol, 1/2 tablet, according to the INR dosage; the patient was administered this medicine on the seventh day of admission). She recovered well.

When discharged, the patient continued to take oral anticoagulants (4 mg acenocoumarol, 1/2 tablet per day, according to the INR dosage on days 7 and 14, and on a monthly basis thereafter), antiepileptic medicines (200 mg carbamazepine, 1/2 tablet three times a day), and painkillers (one metamizole tablet if necessary).

When discharged, the patient was conscious, cooperative, oriented in time and space, without stiffness in the back of the head, had no nystagmus, showed normal ocular motricity, denied diplopia, had possible deglutition for liquid and solid food, displayed symmetric facies, had no movement deficits, had no superficial tactile sensitivity or coordination disorders and showed a cutaneous plantar reflex with bilateral flexion. The images of the MRI can be seen below in Figure 4.

## 3. Discussion

The diagnoses of the cases presented herein were based on the clinical manifestations, neurological objective examinations, imaging investigations, and biological profiles.

The first case described was a 40-year-old female patient, who had and a voluntary abortion. Following the imaging investigations, the diagnosis of sigmoid sinus and right lateral sinus thrombosis of the lower sagittal sinus and right sinus was confirmed, associated with right temporal cortical–subcortical subacute hemorrhage. The following genotypes were identified for the identification of the hereditary thrombophilia profile: heterozygous for V factor mutation *H1299R* (R2), heterozygous for mutation *4G* of *PAI-1*, and haplotype *A1/A2* of *EPCR*.

The second case was a young female patient, on day 6 of the puerperium after caesarian delivery. As the diagnosis was confirmed after the imaging investigations, namely, thrombosis of the left parietal cortical vein associated with ipsilateral superior parietal subcortical venous infarction. The hereditary thrombophilia test indicated the following genetic variants: heterozygous double genotype for the *C677T* and *A1298C* alleles in the *MTHFR* gene, mutant homozygous for the *V34L* allele in *factor XIII* gene, wild type homozygous genotype *5G/5G* of *PAI-1*, and haplotype *A2/A2 (H2/H2)* of *EPCR*. According to the literature, CVST appeared 13 times more often during the puerperium than throughout pregnancy [12].

The patient also had generalized tonic–clonic convulsions. This case demonstrates the critical nature of a wide differential diagnosis in women with postpartum seizures. While eclampsia is the most frequent source of these instances, additional uncommon illnesses can develop throughout the puerperium which necessitate markedly different care strategies. MRI has considerable sensitivity for detecting the existence of cerebral venous thrombosis; therefore, it should be used, if feasible, to aid in diagnosis. In summary, women’s healthcare practitioners can enhance postpartum patient care by evaluating etiologies other than eclampsia and seeking diagnostics that can differentiate them.

During admission, the progression of the two patients presented herein rapidly became favorable, even though both of them presented with intracerebral hemorrhage.

The uniqueness of our cases consisted first of the neurological symptomatology of the patient in the first case, which was significant, as she presented with motor deficits and aphasia. The first patient also presented with subacute cerebral hemorrhage and thrombosis of several sinus areas. In this patient, we also observed mutations associated with cardiovascular disease and thrombophilia. Furthermore, the second patient presented with sensory and motor impairments, and tonic–clonic convulsive spasms and mutations associated with cardiovascular disease and thrombophilia. Imaging showed that the left parietal cortical vein was associated with ipsilateral superior parietal subcortical venous infarction.

Although both patients had procoagulant changes due to genetic mutations in their coagulation gene profiles, neither patient had a pathological episode until the hospitalizations reported on, which shows the extreme importance of pregnancy in the development of cerebral venous thrombosis, and in the case of the first patient, the even greater importance of genital infection and incomplete abortion in the development of cerebral venous thrombosis.

In these two cases, pregnancy-related cerebral venous thrombosis increased the risk of stroke and led to cerebral hemorrhage. Hemorrhagic infarctions are independent indicators of poor prognosis; therefore, early detection and treatment are critical. As the postpartum and post-abortion situations have distinct hemodynamic properties, we hypothesized that cerebral venous thrombosis would appear differently in these phases, as it did in our patients, but the prognosis and outcome were both favorable.

The prognosis of the two patients was favorable, with complete functional recovery. Even though venous cerebral thrombosis has a severe clinical presentation that may endanger the patient’s life, the application of adequate therapeutic measures in due time leads to recovery.

Physiological changes occur throughout pregnancy that encourage the appearance of venous thromboembolism. Increased coagulation factor production and decreased fibrinolytic activity lead to a physiological hypercoagulant state. Additionally, the pressure exerted by the larger uterus decreases blood flow and may potentially induce stasis. Additionally, acute blood loss after childbirth, postpartum infection and extended bed rest all significantly enhance the risk of venous thromboembolism, especially in patients with mutations associated with cardiovascular disease and thrombophilia. In our first patient, another CVT risk factor besides the pregnancy interruption and mutations was the infection for which the patient received treatment with antibiotics.

A solitary prothrombotic factor might not have been a substantial risk factor for CVT in our patients and may not be common. In physiological conditions associated with increased sensitivity to thrombosis, such as pregnancy, puerperium, and abortion, the occurrence of numerous prothrombotic states can turn the situation in favor of thrombosis. The identification of intrinsic prothrombotic conditions can have a considerable influence on the long-term treatment of puerperal CVT, particularly in terms of oral anticoagulant duration, future pregnancies, and the use of estrogen-containing oral contraceptives.

Cerebral venous thrombosis is a very uncommon disorder that may result in cerebral infarction by raising venous and capillary pressure, thereby impairing cerebral perfusion [13]. The condition is caused by a mismatch between the coagulation and anticoagulation systems, which results in venous outflow obstruction and metabolic disorders in brain cells. Increased capillary pressure facilitates hemorrhagic transition, which happens more often in this condition than in other ischemic states.

In the article of Piazza, a case of cerebral venous thrombosis is described, and it resembles the cases presented in this study. The patient was young and came to the hospital for severe headaches on the left side, nausea, photophobia, and phonophobia. Symptoms were reduced by the treatment; thus, the patient was discharged. However, after 12 h, she came back, reporting recurrent severe headaches. The venogram by magnetic resonance revealed thrombosis of the transverse sinus, left sigmoid and internal proximal jugular vein. The patient started the treatment with heparin of low molecular weight; then she took an oral anticoagulant, warfarin. For the hereditary thrombophilia profile, the test described heterozygosity for the mutation of the V Leiden factor [14].

## 4. Conclusions

The coexistence of risk factors in young patients increases the risk of developing cerebral venous thrombosis. Early diagnosis and treatment of cerebral venous thrombosis are critical for a positive outcome with no neurological deficits.

Maternal stroke, characterized as a stroke that occurs during pregnancy or the postpartum period, is becoming a more acknowledged cause of maternal death and disability. Individuals suffering cerebral venous thrombosis must be hospitalized in a stroke unit or neurology clinic, where they should receive triple therapy—antithrombotic therapy; symptomatic therapy; and when necessary, etiologic therapy. These therapies are used identically in young women with sex-specific risk factors and in all other patients with cerebral venous thrombosis.

Despite cerebral venous thrombosis being an uncommon complication of pregnancy, it must be evaluated in differential diagnoses, particularly when a pregnant woman or a woman who has had an abortion presents with an abrupt neurological impairment that raises suspicion of ischemic disease. Delayed diagnosis can be explained by a number of factors, including the disease’s rareness, the disease’s late presentation, the vague symptoms that are typically attributed to pregnancy, the multiple restrictions associated with examining a pregnant woman, avoidance of imaging examinations and the presumably reduced specificity of imaging studies.

While early detection and effective treatment are strongly related to a favorable outcome, the incidence and diversity of pregnancy-related CVT imply that clinicians seem to have little knowledge of its course and that diagnosis is often incorrect or delayed. The features of CVT vary according to the physiological characteristics of pregnant and postpartum women; a significant rate of cerebral infarction is seen in pregnant CVT patients. Neurological symptoms, including headaches and vision loss, along with motor and sensory impairments and seizures, during pregnancy, must alert clinicians to the possibility of CVT.

In summary, oral contraceptive use and pregnancy or puerperium coincide with a significant number of cerebral venous thrombosis cases in young women, often in conjunction with other prothrombotic risk factors, most notably congenital thrombophilia. Although the clinical manifestations, diagnostic testing, and treatment of these cerebral venous thrombosis cases are non-specific, the overall prognosis is favorable, with full recovery occurring in the majority of patients. Future pregnancies are not contraindicated; however, all estrogen-based contraception is permanently contraindicated.

## Figures and Tables

**Figure 1 life-12-00090-f001:**
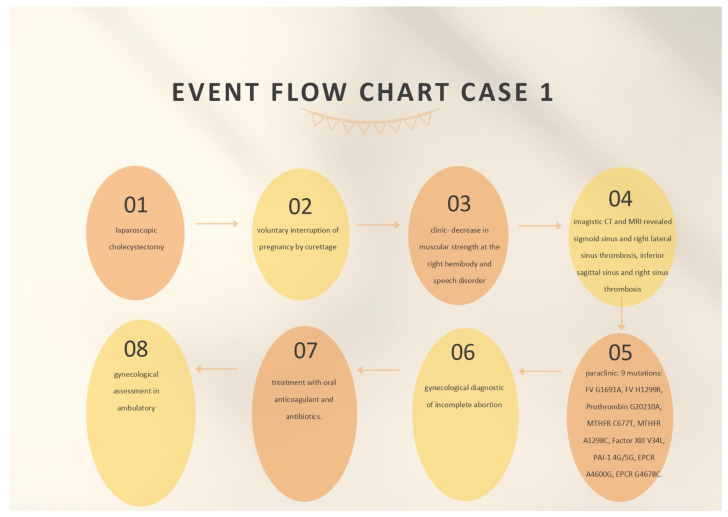
Event flow chart case 1.

**Figure 2 life-12-00090-f002:**
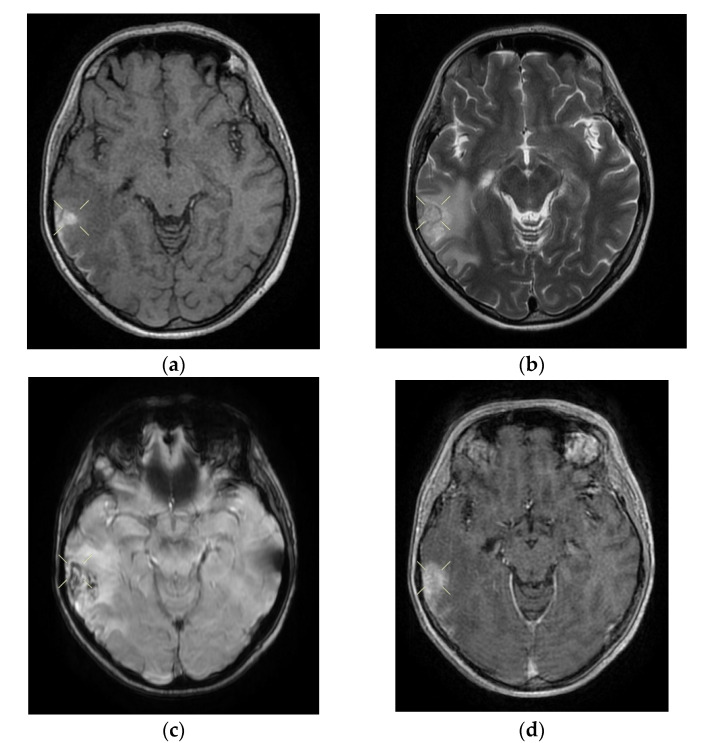
Cerebral native and contrast-enhanced MRI and angiography, and CT cerebral venography highlighting the sigmoid sinus and right lateral sinus thrombosis and the inferior sagittal sinus and right sinus thrombosis, associated with right temporal cortical and subcortical subacute hemorrhage, supratentorial recent subacute synchronous lacunar infarct, (cytotoxic and vasogenic) thalamic–lenticular–caudal edema, and supratentorial non-specific demyelinating lesions. Magnetic resonance imaging shows cortico-subcortical subacute hemorrhage in the right temporal lobe (**a**,**b**) T1 and T2 hyperintensities. (**c**) methemoglobin signal. (**d**) heterogeneous contrast enhancement. (**e**) supratentorial recent subacute lacunar infarction in a millimeter lesion in hypersignal FLAIR, restrictive in diffusion coefficient. (**f**,**g**) supratentorial recent subacute lacunar infarction located in the corpus callosum. (**h**,**i**,**j**,**k**) cytotoxic and vasogenic edema in diffuse T2 and FLAIR high signal and moderate restriction in diffusion coefficient in the left thalamus. (**l**,**m**,**n**,**o**) cytotoxic and vasogenic edema in left lenticular-caudate nucleus. (**p**) right sigmoid and lateral sinuses thrombosis—T1 and T2 hyperintense material, without contrast enhancement. The intravenous post-contrast and native cranio-cerebral MRI examination highlights are as follows: oval globular formations with a non-homogeneous central portion and a periphery with a methemoglobin signal, hyper-intense T1–T2, axial dimensions of 11/10 mm maximally and heterogeneous contrast outlet, along with right temporal cortical and subcortical conglomerates, with extended moderate perilesional oedema; FLAIR hyper-intense millimeter lesions, intense and homogeneous restriction in diffusion and no-contrast outlets in the semioval centers, in the corpus callosum and in the middle temporal gyrus; diffuse signal T2–FLAIR increased in the left and left lenticular–caudal thalamus, with minimum diffusion restriction and no detectable contrast outlets; a few T2–FLAIR hyper-signal millimeter outbreaks, with no diffusion restriction and no corresponding T1, located in the white matter in the periventricular hemisphere and bilateral frontal–parietal subcortical area; normal supra- and infratentorial pericerebral liquid spaces; a symmetric ventricular system, with normal dimensions; structures of the median line in normal position; orbits and orbital content without anomalies; and paranasal sinuses with normal development and pneumatization. Magnetic resonance (MR) cerebral arteriography and venography indicated the following: internal carotid arteries symmetrically disposed, with a normal trajectory and caliber; anterior cerebral arteries and normal average bilaterally detached from the internal carotid, with no areas of stenosis or circumscribed dilation, with a homogeneous intralumenal signal; vertebral arteries, basilar artery, upper cerebral arteries and communicating arteries with a normal trajectory and caliber; hyper-intense T1–T2 material, with a no-contrast outlet, which transversely occupied the sinuses and sigmoid on the right side; and a lesion with the same signal characteristics situated along the right sinus and extended towards the inferior sagittal sinus; the rest of the dural sinuses had no detectable lesions in the sequences observed.

**Figure 3 life-12-00090-f003:**
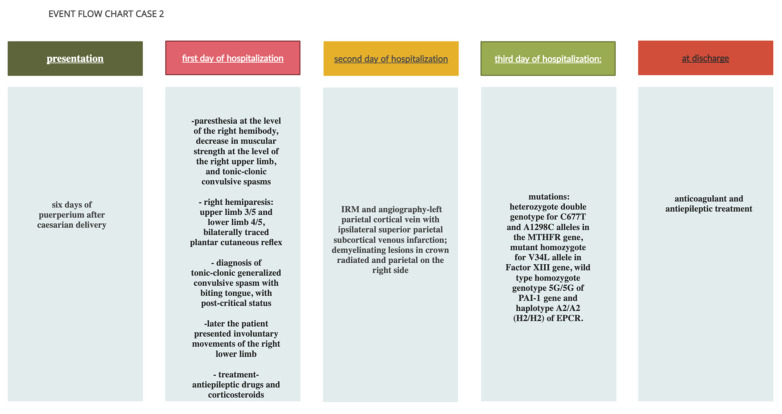
Event flow chart: Case 2.

**Figure 4 life-12-00090-f004:**
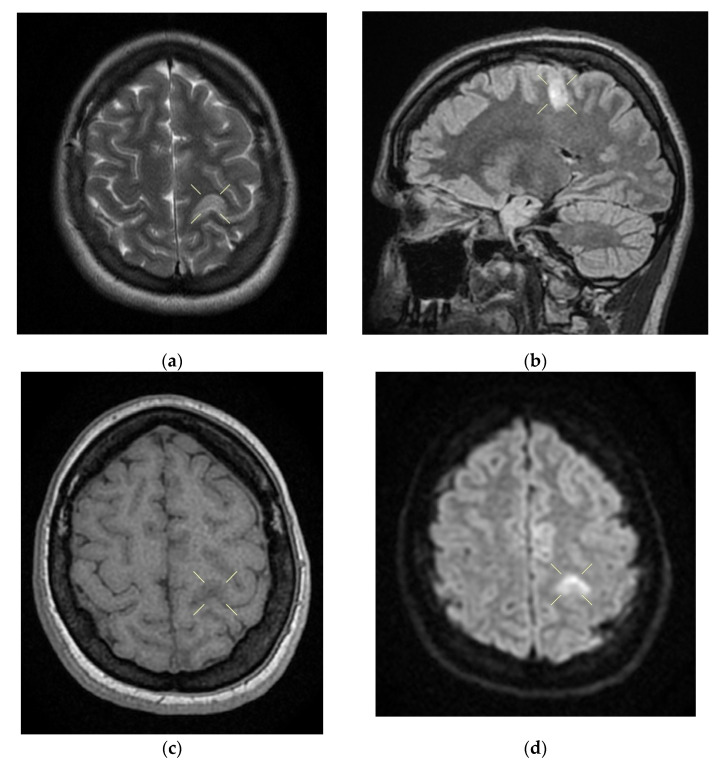
Native and contrast-enhanced cerebral MRI and angiography with arterial and venous sequencing, performed on the second day of hospitalization. The scan indicates thrombosis of the left parietal cortical vein associated with ipsilateral superior parietal subcortical venous infarction—(**a**) band in T2 and FLAIR hypersignal. (**b**) band in T2 and FLAIR hypersignal. (**c**) T1 hyposignal. (**d**) restrictive in diffusion coefficient. (**e**) restrictive in diffusion coefficient. (**f**) restrictive in diffusion coefficient with weak contrast enhancement. Demyelinating lesions are organized in the crown, radiated and parietal on the right side, most probably with an ischemic vascular sublayer. The native cranio-cerebral and post-contrast MRI examination with arterial and venous angiographic sequencing highlighted the following: normal pericerebral liquid spaces; a symmetric ventricular system, with normal dimensions; an area in the hypersignal band T2, and a FLAIR/iso-hypo signal T1, with restricted diffusion weighing, weak gadolinophilia, axial dimensions of about 9/16 mm, located subcortically and parietally on the upper left side; two millimeter focal points of the T2 and FLAIR hypersignal, with no diffusion restrictions or detectable contrast outlet organized in the crown, radiated and parietal subcortical on the right side, in the area adjacent to the dorsal horn of the VL; structures of the median line in the right position; orbits and orbital content without anomalies; paranasal sinuses with normal development and pneumatization; the absence of images evoking hemorrhagic accumulations or masses with a tumor sublayer; symmetrically disposed internal carotid arteries with normal trajectories and caliber; a normal bilateral carotid siphon with no position or extrinsic compression anomalies, with homogeneous intensity of the intralumenal signal; anterior and middle cerebral arteries that were normally detached from the internal carotid on both sides, without any areas of inferior longitudinal stenosis with an aspect within the normal limits; transverse and symmetric sigmoid sinuses, without lesions; the rest of the patient’s evaluable venous segments did not present any defect in the lumen signal.

## Data Availability

Third-party data restrictions apply to the availability of these data. The data were obtained from Constanta’s Sf Apostol Andrei County Emergency Clinical Hospital and are available from the authors with the permission of the Institutional Ethics Committee of Clinical Studies of the Constanta’s Sf Apostol Andrei County Emergency Clinical Hospital.

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
