# Peer review of "Case Reports of Pregnancy-Related Cerebral Venous Thrombosis in the Neurology Department of the Emergency Clinical Hospital in Constanta"

_life, 2022, doi:10.3390/life12010090_

Round 1

Reviewer 1 Report

This is a well designed paper, with clear result and important conclusion.

Minor revision are request for english language.

Author Response

Dear Reviewer,

Thank you for your suggestions, help, and support. 

We have modified the article accordingly to the MDPI English Editing Service instructions that we have received after sending the article to this service. Also, other major modifications have been made to the article.

We hope that we have made the modifications that you are expecting. 

Reviewer 2 Report

Although cerebral venous thrombosis is rare, its occurrence in context during labor and purperium is not uncommon. The clinical vignette description should be brief. There is no proper flow. I would recommend a time flow chart describing events for better readership. The authors should describe generic name of the drug( for eg: Acenocoumarol instead of sintrom).

Author Response

Dear Reviewer,

Thank you for your suggestions, help, and support.

We hope that we have made the modifications that you are expecting.

Although cerebral venous thrombosis is rare, its occurrence in context during labor and purperium is not uncommon. The clinical vignette description should be brief.

There is no proper flow. I would recommend a time flow chart describing events for better readership.

Done. Modifications have been made throughout the entire text of the paper so the description of the events will be easier to read and understand.

Two figures were added with the description of the events: Figure 1 – line 735 and Figure 3 – line 1594.

The authors should describe generic name of the drug( for eg: Acenocoumarol instead of sintrom).

Done. We have made the changes. Lines 736-757 and 1596-1610.

Reviewer 3 Report

The authors presented two classical cases of cerebral venous thrombosis.

I have the the following comments/suggestions:

-the title should be shortened

-in the introduction section should be explained the uniqueness of the presented  cases, and how it contributes to the existing literature.

- please update the basic epidemiology and sex/age distribution by referring to the most recent studies (studies from Netherlands, Australia, Norway, Finland, Romania published in Stroke 2014-2020, Brain Sciences 2021, Journal of Clinical Medicine 2021).

- the case presentation article should not contain materials and methods section

- the two case presentations should be much better structured and shortened focused on the essence

- the laboratory analyzes should not be presented in separate tables, only the relevant analyzes should be  included in the case reports

- too many images are included, only the most suggestive images should be left

- in the discussion chapter it should be specified which were the particularities of the two cases, summarize the key features of the cases and discuss the experiences learnt.

- taking into account that the authors presented two scholarly cases in the conclusions section should be breafly outlined the take-home messages,  suggestions and recommendations to practitioners

Author Response

Dear Reviewer,

Thank you for your suggestions, help, and support.

We hope that we have made the modifications that you are expecting.

The authors presented two classical cases of cerebral venous thrombosis.

I have the the following comments/suggestions:

-the title should be shortened

Done. The title was modified.

-in the introduction section should be explained the uniqueness of the presented  cases, and how it contributes to the existing literature.

Done. We have tried to modify the text in order to underline the uniqueness of the cases presented in this paper. Lines 211-231.

- please update the basic epidemiology and sex/age distribution by referring to the most recent studies (studies from Netherlands, Australia, Norway, Finland, Romania published in Stroke 2014-2020, Brain Sciences 2021, Journal of Clinical Medicine 2021).

Done. We've attempted to change the text in accordance to the most recent studies in the literature. Lines 38-119.

- the case presentation article should not contain materials and methods section

Done. Thank you for the suggestion!

- the two case presentations should be much better structured and shortened focused on the essence

Done. We revised the text's structure by shortening it and emphasizing the most critical points. We underlined the critical points with the addition of two new event charts.

- the laboratory analyzes should not be presented in separate tables, only the relevant analyzes should be  included in the case reports

Done. We have made the change. Thank you!

- too many images are included, only the most suggestive images should be left

Done. Only the most relevant images were chosen.

- in the discussion chapter it should be specified which were the particularities of the two cases, summarize the key features of the cases and discuss the experiences learnt.

Done. We have modified the discussion chapter following your suggestions. Lines 1755-1880.

- taking into account that the authors presented two scholarly cases in the conclusions section should be breafly outlined the take-home messages,  suggestions and recommendations to practitioners

Done. We have improved the conclusions chapter in order to summarize some main findings that would follow your suggestions. Lines 1905-1950.

Round 2

Reviewer 3 Report

I appreciate the authors' intention to improve the article. In order to be acceptable, I consider that further changes are needed:

- The article is still too long for case report and should be reduced in size, the presented  information needs to be better summarized

- The article should be sent to a professional medical english proofreading service

- The introductory chapter is too long, the last 5 paragraphs should be moved and integrated into the discussion chapter

- Tables 1-5 in my view are useless because the relevant information is already included in the text. Tables should be removed from the article

- The two flow charts are incomplete and not informative, for example in flow chart 1 in points 01, 07, 08 there is no text, in point 03 there is the following text “muscular strength at the”, I don't understand what the author wanted to write.

- If the author describes in detail therapeutic options that are not in accordance with the current guidelines, it should be specified why they used the respective drug and the dose (for example: it is a local custom or it was the decision of the treating physician): current guidelines do not recommend routine use of steroid anti-inflammatory drugs in cerebral venous thrombosis; the dose of phenitoin was not specified, except that 1 ampoule was administered.

- The information already described in the present case should not be repeated in the discussion section, only tangentially.

Author Response

Dear Reviewer,

Thank you for your patience, help and suggestions. We hope that we have modified the manuscript accordingly to your suggestions, and we will be willing to make further modifications in order to make it suitable for publication.

I appreciate the authors' intention to improve the article. In order to be acceptable, I consider that further changes are needed:

- The article is still too long for case report and should be reduced in size, the presented  information needs to be better summarized

Done. The results section was summarized by deleting the less important aspects of the cases’ presentations. Also, the introduction and discussion chapters were updated as you suggested (see below).

- The article should be sent to a professional medical english proofreading service

Done. It was reviewed by the experts in the field from the MDPI English Editing Service, and all the modifications that they suggested were made, in order to have a correct manuscript (being suitable for the MDPI journals). If you would recommend new modifications to this matter, we will address another service.

- The introductory chapter is too long, the last 5 paragraphs should be moved and integrated into the discussion chapter

Done. We have moved the last 5 paragraphs into the discussion chapter (line 1711).

- Tables 1-5 in my view are useless because the relevant information is already included in the text. Tables should be removed from the article

Done. We have removed the tables.

- The two flow charts are incomplete and not informative, for example in flow chart 1 in points 01, 07, 08 there is no text, in point 03 there is the following text “muscular strength at the”, I don't understand what the author wanted to write.

Done. The flow charts were upgraded.

- If the author describes in detail therapeutic options that are not in accordance with the current guidelines, it should be specified why they used the respective drug and the dose (for example: it is a local custom or it was the decision of the treating physician): current guidelines do not recommend routine use of steroid anti-inflammatory drugs in cerebral venous thrombosis; the dose of phenitoin was not specified, except that 1 ampoule was administered.

The use of steroid anti-inflammatory drugs in cerebral venous thrombosis was the choice of the treating physician. Phenytoin 250 mg/5 ml was used, the text was modified (lines 1079 and 1084).

- The information already described in the present case should not be repeated in the discussion section, only tangentially.

Done. The discussion section was modified accordingly.
